# MIXTURE OF NEURAL OPERATOR EXPERTS FOR LEARNING BOUNDARY CONDITIONS AND MODEL SELECTION

## ABSTRACT

While Fourier-based neural operators are best suited to learning mappings between functions on periodic domains, several works have introduced techniques for incorporating non trivial boundary conditions. However, all previously introduced methods have restrictions that limit their applicability. In this work, we introduce an alternative approach to imposing boundary conditions inspired by volume penalization from numerical methods and Mixture of Experts (MoE) from machine learning. By introducing competing experts, the approach additionally allows for model selection. To demonstrate the method, we combine a spatially conditioned MoE with the Fourier based, Modal Operator Regression for Physics (MOR-Physics) neural operator and recover a nonlinear operator on a nontrivial 2d domain. Next, we extract a large eddy simulation (LES) model from direct numerical simulation of channel flow and show the domain decomposition provided by our approach. Finally, we train our LES model with Bayesian variational inference and obtain posterior predictive samples of flow far past the DNS simulation time horizon.

## 1 INTRODUCTION

Fully resolved simulations of partial differential equations (PDEs) are prohibitively expensive for most systems, even on the largest supercomputers. Accordingly, under-resolved simulations are supplemented with models for subgrid-scale dynamics. For example, for the Navier-Stokes equations in the turbulent regime, the turbulence dynamics are approximated statistically via large eddy simulation (LES) or Reynolds averaged Navier-Stokes (RANS) models; such models are imperfect as they encode various empirical assumptions and hand-tuned approximations Pope (2000).

Neural operators are a class of surrogate models that parameterize unknown operators with neural networks and can learn PDE models from high-fidelity simulation and/or experimental data. Originally introduced as a theoretical construction by Chen & Chen (1995), recent numerical implementations of neural operators have seen recent success in learning a variety of PDEs, e.g., Patel & Desjardins (2018); Patel et al. (2021); Li et al. (2021); Lu et al. (2021); Rahman et al. (2023); Tripura & Chakraborty (2023b).

Neural operators utilizing the Fourier transform, e.g., MOR-Physics Patel & Desjardins (2018) and Fourier Neural Operator (FNO) Li et al. (2021) are particularly attractive due to the simplicity of parametrization and computational efficiency. However, the Fourier transform exhibits Gibbs phenomenon when attempting to model discontinuities and cannot handle non-periodic boundary conditions. Fourier Neural Operators (FNOs) Li et al. (2021) tries to remedy this by adding "bias functions", $Wv(x)$ that operate along the Fourier convolution layers. However, Lu et al. (2022) has shown that this does not fully alleviate the weakness that FNO has around non-periodic boundary conditions and discontinuities, e.g., shock waves. Tripura & Chakraborty (2023a) introduced wavelet neural operators (WNO) to address limitations of Fourier based neural operators for problems involving neural operators, arguing that wavelet bases contain both spatial and spectral information and can better resolve discontinuities and spikes and circumvent the Gibbs phenomenon. Another neural operator, NORM Chen, Gengxiang et al. (2024), generalizes the modal strategy to Riemannian manifolds, but requires precomputation of the eigenfunctions of the Laplace-Beltrami

operator and lacks a fast transform. (Li et al., 2023), learns in addition to an FNO, a deformation from a nontrivial domain to a periodic domain but is ill-suited for higher dimensional domains Chen, Gengxiang et al. (2024).

Instead, we address model discontinuities and boundary conditions by proposing a mixture of experts model, where the weighting of experts is determined locally across space via a partition of unity (POU). This mechanism enables the model to choose different experts on each side of a discontinuity or non-periodic transition, e.g. zero velocity at the walls for channel flow. We apply a POU-Net Lee et al. (2021); Shazeer et al. (2017) with a set of MOR-Physics Operators Patel & Desjardins (2018) to better learn boundary conditions, introducing POU-MOR-Physics. In addition to being well suited to nontrivial boundary conditions, our method interpretably divides the domains for the experts. To demonstrate the method, we simultaneously learn the boundary conditions and a LES model for channel flow using the Johns Hopkins Turbulence Database (JHTDB) Li et al. (2008); Graham et al. (2016); Perlman et al. (2007). We find this approach to demonstrate superior performance over WNO for LES modeling.

We also take measures to accommodate the limited high-fidelity JHTDB DNS dataset. We embed a forward-Euler PDE solver in our model for training and implement uncertainty quantification (UQ) to provide a range of predictions outside the training data. The integration of the PDE solver with the learned correction serves a purpose similar to data augmentation: enabling higher-quality predictions despite the limited data. The UQ is implemented with Mean-Field Variational Inference (MFVI) Blei et al. (2017), which enables (1) the identification of the support of the dataset at prediction time (and hence detection of out-of-distribution (OOD) queries), and (2) the modeling of the notoriously high aleatoric uncertainty that are inherent to turbulence.

We leverage our neural operator to provide the missing physics for the subgrid scale dynamics in turbulent channel flow, i.e., LES closure modeling. To the best of our knowledge, the current state of the art for LES modeling of wall bounded turbulence with operator learning was presented by (Wang et al., 2024). The authors used an UNet enhanced FNO to learn a model for flows up to Re=590. In comparison, with the novel methods presented in this paper, our LES model incorporates UQ, *a priori* known physics, models flows up to Re=1000, and is more interpretable due to Mixture-of-Expert style partitioning.

## 1.1 CONTRIBUTIONS

In this work we,

- Describe a connection between volume penalized numerical methods and POU-Nets.
- Leverage the connection to develop a novel operator learning strategy capable of learning multi-physics systems with non-trivial boundary conditions.
- Demonstrate the method in simple 2D operator learning problems.
- Advance the State-of-the-Art relative to Wang et al. (2024), by accurately modeling Re=1000 3d wall bounded turbulence via neural operators and quantifying uncertainties.
- Outperform WaveletNO and CNN SATO comparison models on the LES model learning task.

## 2 METHODS

Given pairs of functions as data,

$$D = \{(u_i, v_i) | i \in 1, \ldots, n, u_i \in U, v_i \in V\},$$

where $U$ and $V$ are two Banach spaces, we seek an operator, $\mathcal{P} : U \to V$. The spaces $U$ and $V$ are sets of functions defined on $u_i : X \to \mathbb{R}^m$ and $v_i : Y \to \mathbb{R}^p$, where $X$ and $Y$ are compact subsets of $\mathbb{R}^{d_1}$ and $\mathbb{R}^{d_2}$, respectively. Given our target application, we will consider $X = Y$ and therefore $d_1 = d_2 = d$. Operator learning introduces a parametrization for the unknown operator, $\mathcal{P} : U \times \Phi \to V$, where $\phi \in \Phi$ are parameters. As we demonstrate with learning an LES closure model in Section 3.2, models can also incorporate *a priori* known physics.

Given the parametrization, we can learn a deterministic model by solving the minimization problem,

$$\phi = \underset{\hat{\phi}}{\operatorname{argmin}}\ L(D, \mathcal{P}(\cdot, \hat{\phi})), \tag{1}$$

where $L$ is a loss function, e.g., least squares. Alternatively, we can learn a probabilistic model by solving the optimization problem,

$$Q = \underset{\hat{Q}}{\operatorname{argmin}}\ L'(D, \mathcal{P}, \hat{Q}), \tag{2}$$

where $Q$ is a variational distribution over $\Phi$ and $L'$ is a loss function, e.g., the negative of evidence lower bound (ELBO). The recovered operator must respect boundary conditions, which for neural operators leveraging the Fourier transform is a nontrivial task Li et al. (2023).

One approach, volume penalization Brown-Dymkoski et al. (2014); Kadoch et al. (2012), is an embedded boundary method that integrates a variety of PDEs with complex boundary conditions. Volume penalization partitions a simple domain in two, where one of the subdomains is $X$, and applies forcing in each subdomain such that the solution to the new PDE, when restricted to $X$, approximates the solution to the original PDE. More generally, different kinds of physics may operate in different regions of space, and must be modeled as a collection of PDEs in disjoint domains, e.g., fluid-structure interaction Engels et al. (2015). For these multi-physics problems, volume penalization partitions the extended domain accordingly with the appropriate forcing. In this work, we focus on Fourier pseudo-spectral methods which, while simple, efficient, and accurate, only solve PDEs on periodic domains with Cartesian meshes in their base form. Applied to Fourier pseudo-spectral methods Kolomenskiy & Schneider (2009), volume penalization retains the simplicity of the original discretization while expanding its reach to nontrivial problems.

Our approach identifies physical systems with complex boundary conditions or multiphysics systems by learning the requisite partitions and forcing in a volume penalized Fourier pseudo-spectral scheme. We approach this task with the machine learning technique, mixture of experts, where experts learn their operations in subdomains of the problem space partitioned by learnable gating functions.

Taking inspiration from volume penalization and mixture of experts, we parameterize $\mathcal{P}$ using a mixture of neural operators where each neural operator is a parameterized Fourier pseudo-spectral operator. This composite neural operator, which we refer to as POU-MOR-Physics, however, relies on the Fourier transform, so is designed for smooth functions with periodic domains. Ideally, a smooth extension for $u_i$ to this periodic domain is constructed to avoid Gibbs phenomena originating from the interface between $X$ and the periodic domain. We first discuss a strategy for constructing a smooth extension of $u_i$ in the next section that is compatible with the neural operator in Section 2.2.

### 2.1 Feature Engineering – Smooth Extension of Functions to Periodic Domains

Our neural operator, $\mathcal{P}$, discussed in Section 2.2, is parameterized via the Fourier transform and therefore is only well-defined for periodic functions. However, $U$ and $V$ are functions on the torus, $\mathbb{T}^d$, so we embed $X$ in $\mathbb{T}^d$, and must supply extensions for our functions in the new domain. Our complete prediction mechanism, including the neural operator, restriction, and extension, takes the form,

$$\mathcal{P}_e : u \overset{\mathcal{E}}{\mapsto} u_e \overset{\mathcal{P}}{\mapsto} v_e \overset{\mathcal{R}}{\mapsto} v. \tag{3}$$

where $\mathcal{E}$ is an extension and $\mathcal{R}$ is a restriction; we have suppressed dependence on the parameters, $\phi$,

Since we only compute losses in $X$, we do not choose an extension for the output functions and allow our neural operators to produce actions (i.e. $v \in V$) that behave arbitrarily in the complement, $\mathbb{T}^d \setminus X$. Our procedure provides a smooth extension of input functions to the torus. Input functions with constant traces on the boundary are extended simply by setting the function value in $\mathbb{T}^d \setminus X$ to the constant value. For other functions, a similar approach would lead to discontinuities, which induce Gibbs phenomena due to the Fourier transforms in our parametrization. We discuss the details of our approach to extension in Appendix C. For the remainder, we will drop the subscript, $e$, and identify $u_e$ with $u$, $v_e$ with $v$, and $\mathcal{P}_e$ with $\mathcal{P}$.

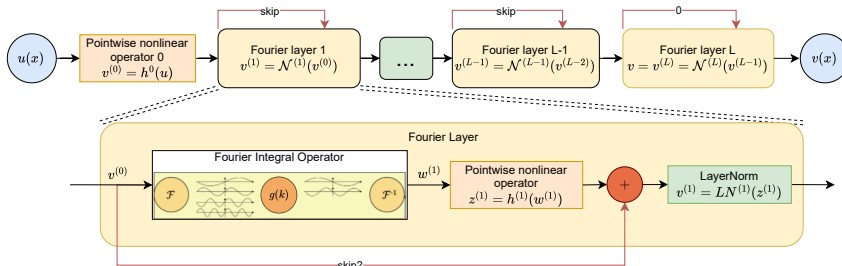

Figure 1: MOR Operator diagram, showing architecture of each expert operator $\mathcal{N}_i$. Black arrows denote function composition.

## 2.2 MODEL DESIGN – PARTITION OF UNITY NETWORK WITH MOR-PHYSICS NEURAL OPERATORS

We now construct a neural operator, POU-MOR-Physics, for functions with domain, $\mathbb{T}^d$. Our parameterization is a spatially conditioned POU-network composed of a gating network and neural operator experts:

$$(\mathcal{P}(u;\phi))(x) = \sum_{i=1}^{I} G_i(x;\phi_{G_i})(\mathcal{N}_i(u;\phi_{\mathcal{N}_i}))(x), \tag{4}$$

where $G$ forms a partition of unity, i.e. $\forall x \in \mathbb{T}^d$,

$$\sum_{i}^{I} G_i(x;\phi_{G_i}) = 1 \quad \text{and} \quad G_i(x;\phi_{G_i}) \geq 0, \tag{5}$$

and where $\phi$ are the combined set of parameters in the full model and $\phi_*$ are parameters for the various subcomponents of the model. Where the context is clear, we will suppress in our notation the dependence on the parameters. We describe each of the components of $\mathcal{P}$ – the neural operators $\mathcal{N}_i$, $i = 1, \ldots, I$, the gating network, $G$, and the time-dependent autoregressive strategy – in turn, which then situates the model to be used for mean-field variational inference (MFVI).

### 2.2.1 MOR-PHYSICS OPERATOR

The neural operators in Equation equation 4, $\mathcal{N}_i$, are modified versions of the MOR-Physics Operator presented in Patel & Desjardins (2018); Patel et al. (2021). For convenience, we drop the subscript $i$, as we define each $\mathcal{N}_i$ operator similarly. We will exclusively parameterize operators with the approach detailed below. However, if we have *a priori* knowledge about a system, we can use a predefined operator in addition to the learned ones. In our exemplars, we include the zero expert in our ensemble of experts, i.e., an operator that evaluates to zero for any input function.

We compose $L$ MOR-Physics operators, $\mathcal{N}^{(l)}$, $l = 1 : L$, thereby introducing latent functions, $v^{(l)} : \mathbb{T}^d \to \mathbb{R}^{m^{(l)}}$, after the action of each operator,

$$v^{(l+1)} = \mathcal{N}^{(l+1)}(v^{(l)}) = LN\left[v^{(l)} + h^{(l+1)} \circ \mathcal{F}^{-1}(g^{(l+1)} \cdot \mathcal{F}(v^{(l)}))\right], \tag{6}$$

where $\mathcal{F}$ is the Fourier transform, $h^{(l)}$ is a learned point-wise operator, implemented with a neural network, $g^{(l)}$ is a weighting function in Fourier frequency space, and $LN$ is layer normalization. The skip connection in the above equation can also be removed. The multiplication operation, $\cdot$, in Equation equation 6 is both the Hadamard product and a matrix multiplication, i.e., at every wavenumber, $k$, we have a matrix-vector product between $g^{(l+1)}(k) \in \mathbb{C}^{m^{(l+1)} \times m^{(l)}}$ and $\mathcal{F}(v^{(l)})(k) \in \mathbb{C}^{m^{(l)}}$. This operation is comparable to linear operations across channels in a convolutional neural network (CNN). .

To build these operators, we discretize the domains on Cartesian meshes and replace the abstract linear operations with the appropriate matrix operations. Depending on the problem, we implement $g^{(l)}(k)$ either as a neural network or a parameterized tensor. In either case, we truncate the higher

modes of either the $g^{(l)}(k)$ or $\mathcal{F}(v^{(l)})(k)$ tensors (which ever has more in a given dimension) so that their shapes are made compatible. This effectively results in a low pass filter; more details can be seen in Patel & Desjardins (2018).

Ultimately, each $\mathcal{N}_i$ is then built via composition of several layers $\mathcal{N}^{(l)}$, so that

$$v = v^{(L)} = \mathcal{N}_i(u) := (\mathcal{N}^{(L)} \circ \mathcal{N}^{(L-1)} \circ \cdots \circ \mathcal{N}^{(1)} \circ h^0)(u).$$

and $v^{(L)} = v$ provides the action of the composite MOR-Physics operator. This architecture is sketched in Figure 1, mapping from an input $u$ to a target $v$ through the intermediate layers $v^{(l)}$. As required to allow for $v^{(l)}$ to vary in output dimension, the skip connections in equation 6 can be removed.

### 2.2.2 GATING NETWORK

The gating network $G : \mathbb{T}^d \to \mathbb{R}^I$ is built from a neural network, taking the coordinates as inputs. Its output logits are transformed via softmax or a custom designed importance normalized softmax (see Appendix D for details) into coefficients to compute a convex combination of neural experts, $\mathcal{N}_i$, at location $x \in \mathbb{T}^d$. The Gating network does *not* take $u(x)$ as input, it only uses the location $x$ which is sufficient to partition the space for different experts. See Figure 2 for a schematic of the gating function.

Since we rely on the Fourier transform for our operator parameterization, we use a smooth mapping, $\mathbb{T}^d \to \mathbb{R}^{2d}$, to provide the input to the gating network. The domain, $\mathbb{T}^d$, is parameterized by $d$ angles, $\theta_i$ and mapped to a vector in $\mathbb{R}^{2d}$ as,

$$[\sin(\theta_1), \ldots, \sin(\theta_d), \cos(\theta_1), \ldots, \cos(\theta_d)]^T. \tag{7}$$

This yields more consistently interpretable expert partitions; In our 2D exemplar discussed in Section 3.1 we found this approach to produce symmetric partitions that conform to the problem specification, while the simpler approach using the angles as coordinates led to asymmetric partitions.

Although we describe a neural network gating function in this section, *a priori* known gates can replace the network, e.g., when a PDE domain is already well characterized.

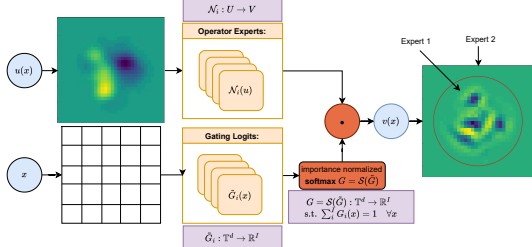

Figure 2: Mixture of Experts model, where a weighted sum of gating weights $G_i(x)$ is applied to expert outputs $\mathcal{N}_i(u)$. The gating weights are spatially localized, depending only on $x$ and not $u$.

### 2.2.3 AUTOREGRESSIVE POU-MOR-PHYSICS MODEL

Autoregressive models are well-suited for learning spatiotemporal dynamics. We can specialize the model in equation 4 to learn an update operator representing the evolution of the system over a small period of time, $\Delta t$, by letting $\mathcal{P} : \mathcal{U} \to \mathcal{U}$ and,

$$\begin{aligned} u(\cdot, (n+1)\Delta t) &= \mathcal{P}u(\cdot, t) \\ &= u_{n+1} = \mathcal{P}u_n. \end{aligned} \tag{8}$$

In this context, the operator $\mathcal{P}$ can be composed with itself to predict the system at discrete times, $u_{n+p} = \mathcal{P}^p u_n$. In many cases, parts of a model are *a priori* known and we have a PDE with an unknown operator,

$$\partial_t u = \mathcal{M}u + \tilde{\mathcal{P}}u \tag{9}$$

where $\mathcal{M}$ is known and $\tilde{\mathcal{P}}$ is unknown. A first order in time operator splitting allows for the update,

$$u_{n+p} = [\mathcal{P}(I + \Delta t\mathcal{M})]^p u_n = (\mathcal{P} \circ \mathcal{E})^p(u_n) = U^p u_n \tag{10}$$

where $\mathcal{P}$ is a new unknown operator that provides the same effect as $\tilde{\mathcal{P}}$. Since the Euler update operator, $\mathcal{E} = I + \Delta t\mathcal{M}$, is a common time integrator for PDEs, we will refer to it as the PDE solver for the remainder of this work. We will demonstrate an autoregressive model in Section 3.2 by learning an LES closure model for wall bounded turbulent flow.

We introduce the operator, $\overline{\mathcal{U}}^p$, to give a time series prediction of $p$ time steps from an initial field, and denote its action on an initial condition, $\overline{\mathcal{U}}^p u_n = u_{[n,n+p]}$, where the subscript indicates a time-series beginning from timestep $n$ and ending with timestep $n + p$.

Given a time-series of functional data, $D = \{\tilde{u}_n\}^{n=0,\dots,N}$, an autoregressive model can be found by solving an optimization problem, i.e., equation 1 or equation 2.

## 2.3 MEAN-FIELD VARIATIONAL INFERENCE (MFVI)

The auto-regressive nature of our Bayesian MFVI model (in Section 3.2) necessarily changes our otherwise standard implementation of MFVI. The details of the major difference are highlighted in Figure 3.

To build MFVI into our neural operator, we assume independent Gaussian variational posteriors for each model parameter, exploiting the conventional reparameterization trick Kingma (2013) for normally distributed weights; see Appendix A.1 for how it applies to complex parameters. Further details on the VI method can be found in Blundell et al. (2015).

Following Blundell et al. (2015)'s convention we treat the VI model as a probabilistic model, in the sense that it directly predicts the mean and variance of Gaussian likelihoods, that is $\mathcal{P}_{LES} : U \to U$, where $U = M \times \Sigma$, such that the mean and variance are given by $u = (\mu, \sigma) \in U$. Only the $\mu$ fields are integrated by the PDE solver as described in Figure 3. We let $\mathcal{P}$ be a neural operator producing unconstrained real vector valued functions, $\mathcal{P}(\mathcal{E}(\mu^{(n)}), \sigma^{(n)}) : x \mapsto (\nu^{(n+1)}, \rho^{(n+1)})$, and constrain its outputs with the following transformations,

$$\mu(\nu) = 2 * \tanh(\nu) \quad \sigma(\rho) = \log(1 + e^\rho) + \epsilon, \tag{11}$$

We refer to the composition of these constraint transformations, the neural operator, and the *a priori* known physics as $\mathcal{P}_{LES}$.

Given a time-series of data, $\tilde{v}$, we learn a Bayesian model using variational inference by optimizing the ELBO,

$$\max_q \left(\log p\left(\tilde{v}_{[0,N]} | \overline{\mathcal{U}}^P \tilde{u}_0\right) - D_{KL}(q||p_0)\right) \tag{12}$$

where $q$ is a variational distribution over $\Theta$, $p_0$ is a prior distribution over $\Theta$, $\log p$ is a Gaussian log likelihood with mean and variance provided by $u$. We use MFVI and select $q$ to be a diagonal Gaussian variational distributions with softplus positivity constraints for the standard deviations Blundell et al. (2015). Equation equation 12 is not computationally tractable, so we use a strided sliding window strategy and obtain,

$$\max_q \sum_{m=0:N:P} \left(\log p\left(\tilde{v}_{[m,m+P]} | \overline{\mathcal{U}}^P \tilde{u}_m\right) - D_{KL}(q||p_0)\right) \tag{13}$$

where $\tilde{u}_{[n,n+P]}$ is the subset of data from timestep $n$ to $n + P$ and $P$ is a hyperparameter for the number of autoregressive steps used during training. The notation, $0 : N : P$, borrows from Python's array slicing syntax. To initialize the means and variances, we set, $\forall m, \tilde{u}_m = (\tilde{v}_m, \tilde{\sigma}_m = 0)$. We approximate the sum via stochastic mini-batch optimization. Overlapping sliding windows would constitute data-augmentation, which has been shown to cause the cold posterior effect (CPE) Izmailov et al. (2021), therefore we avoid using them.

With MFVI, the resulting model outputs Gaussian distribution predictions by applying a weighted sum of the experts' output tensors, c.f. Equation equation 4, where now these output tensors contain both $\mu$ and $\sigma$ channels for each output feature.

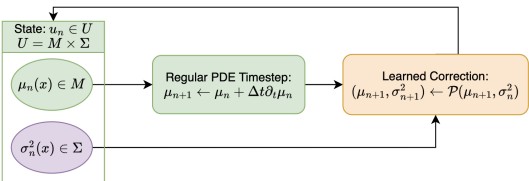

Figure 3: Flow of Uncertainty through Learned PDE solver model. The PDE solver operates on the mean field, $\mu_n \in M$, while the uncertainty field, $\sigma_n \in \Sigma$ is entirely modeled and updated by the Neural Operator.

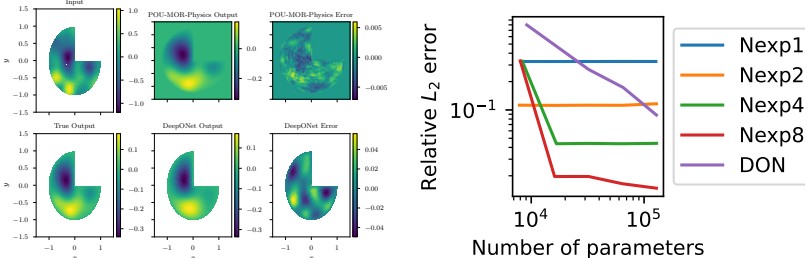

Figure 4: *(Left)* Visual comparison of POU-MOR-Physics (8 experts) and DeepONet learned operators for nonlinear Poisson equation with nontrivial boundary conditions. Both architectures contain $\sim 100K$ parameters. *(Right)* relative errors for POU-MOR operator with varying number of parameters and experts and DeepONet for varying number of parameters. Nexp$I$ refers to POU-MOR-Physics with $I$ experts. DON refers to DeepONet

## 3 NUMERICAL DEMONSTRATIONS

We demonstrate our method with two numerical examples. This section describes the problem formulations and our results. In first example, we learn the solution operator to a nonlinear Poisson equation with mixed boundary condition. In the second, we learn an LES model for turbulent channel flow.

### 3.1 SOLUTION OPERATOR FOR NONLINEAR POISSON PROBLEM ON THREE-QUARTERS DISK

In this section, we demonstrate the mixture-of-experts in POU-MOR-Physics provides a viable approach to learning solution operators for nonlinear PDEs with complex domains. We begin by generating pairs of data, $(u_i, v_i)$, that solve a nonlinear Poisson equation,

$$\begin{aligned} \nabla \cdot \tanh(\nabla v) = u & \quad x \in \Omega \\ v = 0 & \quad x \in \partial\Omega \end{aligned} \tag{14}$$

where the domain, $\Omega$, is the three-quarters of the unit disk shown in Figure 4. We generate $u_i$ by sampling a Gaussian process with mean zero and square exponential covariance kernal with correlation length, $l = 0.2$. For each $u_i$ we solve equation 14 for $v_i$ using a finite element code with P1 elements and a characteristic mesh size of 0.025. We generate 10000 samples of $(u_i, v_i)$, partitioned by 60%/20%/20% into training, validation, and test sets. We train solution operators, $\mathcal{P} : u_i \mapsto v_i$ using POU-MOR-Physics, varying the parameter count and number of experts, and DeepONet for similar parameters counts. The left subplot in Figure 4 shows a visual comparison between actions recovered for POU-MOR-Physics and DeepONet for around 100K parameters. We observe that POU-MOR-Physics has substantially lower error than DeepONet for this problem. On the right subplot, we plot the average relative $L_2$ error over the test set between the FEM solution and the neural operators. We find that the experts are necessary to recover the solution operator and can further improve accuracy as more experts are added. With enough experts, POU-MOR-Physics outperforms DeepONet for all parameter counts tested. DeepONet appears to converge somewhat faster but even if there is a parameter count for which DeepONet performs better, it is likely to be unreasonably high number of parameters to be competitive with POU-MOR-Physics. We provide training and parameterization details in Appendix E.

| Model: | ELBO: val ↑ | ELBO-long ↑ | $R^2$: val ↑ | $R^2$-long ↑ | MAE: val ↓ | MAE-long ↓ |
|---|---|---|---|---|---|---|
| Recursive Steps | 5.47153 | 5.2897 | 99.081% | 98.864% | 0.001802 | 0.0019609 |
| MoE+IN | 5.16139 | -93.94615 | 98.431% | 24.262% | 0.0026547 | 0.02711 |
| MoE | 5.06351 | -671.67914 | 98.505% | -699.483% | 0.0030291 | 0.080844 |
| Control | 5.12027 | -233.91377 | 98.743% | -488.797 % | 0.0029808 | 0.079651 |
| Ours | **5.63422** | **5.48693** | **0.99456%** | **0.99325%** | **0.0015041** | **0.0016127** |

Table 1: Ablation Study (Appendix I): Shorthand: IN=Importance Normalization, LN=LayerNorm, MoE=Mixture of Experts, PP-LL=Posterior-Predictive-Log-Likelihood. Ours combines all features described. Recursive Steps has 10 auto-regressive timesteps (as does our model).

| Model: | ELBO: val ↑ | ELBO: 100 step ↑ | $R^2$: val ↑ | $R^2$: 100-step ↑ | MAE: val ↓ | MAE: 100 step ↓ |
|---|---|---|---|---|---|---|
| CNN-k4 | 3.41832 | 2.9057 | 0.24721 | 0.23969 | 0.022192 | 0.027813 |
| Ours-k4 | **3.54017** | **3.44403** | **0.37666** | **0.35351** | **0.017363** | **0.018602** |

Table 2: CNN Comparison (CNO Proxy: Ours>CNN≥CNO): we make the argument in subsection I.2 that our CNN model serves as a band-unlimited proxy to CNO (similar to a bandlimited CNN).

## 3.2 LARGE EDDY SIMULATION MODELING

Our target application is extracting an LES model from 3d+1 DNS provided by the JHTDB dataset Graham et al. (2016). The simulation data is obtained from the Re=1000 channel flow problem with no-slip boundary conditions (BCs) on the top and bottom of the flow, and periodic BCs on the left, right, front and back. See Appendix H for these details.

We subsample the DNS data spatially (see Table 4 in Appendix 4) and keep the full resolution in the time dimension, motivated by the need to have a larger training dataset. The spatial sub-sampling is performed after applying a box filter to the DNS data, ensuring the sub-sampled grid is representative of the whole DNS field.

From this data, we seek to obtain an autoregressive (Section 2.2.3), MFVI (Section 2.3) model that incorporates physics by leveraging the standard LES formulation (Appendix F). While the JHTDB dataset only simulates the flow for one channel flow through time, $t = T$, we use our model to extrapolate the flow to long times, $t = 2T$, and evaluate the predictions and uncertainties from our model.

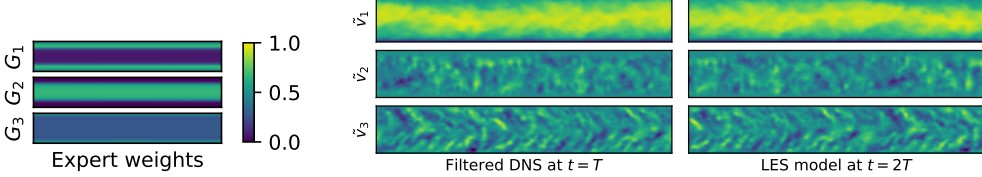

Figure 5: *(Left)* 3D Expert Partitions for JHTDB dataset. $G_3$ is the zero expert, i.e., $\mathcal{N}_3 = 0$. Fields from filtered *(Center)* DNS at $t = T$ and learned *(Right)* Sample of LES model predictions after 2 channel flow-through times. We observe excellent agreement between the learned model and the filtered DNS data, and we find the partitions to align with the channel walls.

| Model: | ELBO: val ↑ | ELBO: 100 step ↑ | $R^2$: val ↑ | $R^2$: 100-step ↑ | MAE: val ↓ | MAE: 100 step ↓ |
|---|---|---|---|---|---|---|
| WNO | 1.91353 | 2.16997 | -3.43538 | -1.78457 | 0.036095 | 0.031982 |
| Ours | **2.95503** | **2.82594** | **0.47534** | **0.34528** | **0.0172** | **0.018449** |

Table 3: WNO Comparison: see subsection I.1

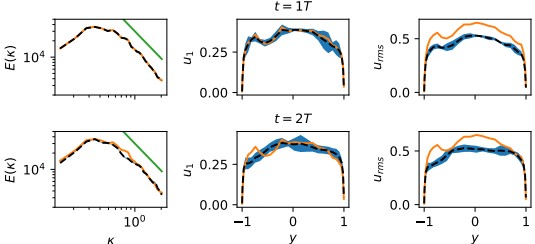

Figure 6: MFVI LES model captures *(Left)* energy spectrum and *(Center)* bulk velocity, and *(Right)* RMS fluctuations. Posterior predictive samples are shown in black & uncertainty bars in blue. The DNS results are shown in orange. Kolmogorov's 5/3 rule is shown in green.

We use our LES model to evolve the system to 2 channel flow-through times. Since the JHTDB only includes one flow length's worth of data, we are predicting dynamics in an extrapolatory regime and using the VI model to provide predictions with error bars. In Figure 5, we find that the gating network is able to partition the domain and find separate models for the bulk and boundary layers of the JHTDB channel data. We see spatial partitions that reflect the boundary layer, with different support for $\mathcal{N}_1$ vs. $\mathcal{N}_2$ Moreover, the transition from $\mathcal{N}_1$ to $\mathcal{N}_2$ is continuous roughly approximating the strength of the velocity at those points in the simulation. This demonstrates the model's ability to find multiple, spatially conditioned models. In Figure 5, we also show a comparison between the filtered DNS fields at $t = T$ and the model predictions at $t = 2T$, observing qualitatively similar fields.

Since the flow is at a statistical steady state, we should see the same mean statistics for our model as the DNS data. Figure 6 shows a statistical comparison between our model and the filtered DNS data. We see that the energy spectrum closely matches that of DNS and obey's Kologmorov's 3/5's rule (the error bars don't pass the threshold). In the middle figures we see close agreement also even at t=2T. However we also see the uncertainty is as expected increasing with time. Finally we see decent agreement with RMS velocity fluctuations in the right most figures, however there is still some room for improvement as matching RMS fluctuations Pope (2000) is more difficult. We demonstrate in Appendix B that the velocities predicted by the POU-MOR-Physics model closely matches the filtered DNS.

We perform an ablation study for POU-MOR-Physics, considering various components of our model (see Table 5). We find that the mixture-of-experts and the recursive timestepping to be essential for our approach. In Table 7 and Table 6 we find our approach to outperform equivalent CNN and a WNO models, respectively. See Appendix **??** for details on these studies.

## 4 CONCLUSION

In this work, we addressed the challenge of learning non-periodic boundary conditions and discontinuities in PDE simulations using a Mixture of Experts (MoE) approach, specifically applied to the MOR-Physics neural operator. The motivation for this work stems from the limitations of traditional Fourier-based PDE solvers, which struggle with non-periodic boundary conditions, such as those found in Navier-Stokes simulations. By integrating a forward Euler PDE solver and applying a correction operator at each timestep, we were able to enhance solver accuracy. Our methods were tested on both 2D synthetic data and LES for 3D channel flow, demonstrating promising results. The model was highly successful in the 2D case, where it flawlessly captured the zero boundary conditions. Furthermore, the Bayesian 3D channel flow problem yielded promising results with a close match between the energy spectra of the DNS and model predictions.

## 5 REPRODUCIBILITY STATEMENT

If accepted, we will provide a github repository for the 2D Poisson example and the LES turbulence model. We will provide methods for constructing, training, and evaluating the models. We refer readers to the JHTDB Graham et al. (2016) for the turbulent channel flow dataset. We will

provide code for generating synthetic data for the 2D Poisson example. This code cannot easily be anonymized due to references to specific computational infrastructure, so we cannot provide it during the review period.

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

## A  APPENDIX

### A.1  COMPLEX VALUES AND REPARAMETERIZATION TRICK

The complex-valued weights in the network architecture in Equation equation 6 are defined as $\theta_i = w_i + i\tilde{w}_i$, where $\theta_i \in \mathbb{C}$, and the values $w_i \sim N(\mu_i, \sigma_i^2), \tilde{w}_i \sim N(\tilde{\mu}_i, \tilde{\sigma}_i^2)$, with Gaussian values parameterized by the reparameterization trick. This scheme yields twice as many parameters (1/2 real, and 1/2 imaginary) as compared to a comparable network parameterized with real-valued weights.

## B  FILTERED DNS AND LES FIELDS FROM DETERMINISTIC MODEL

We include here a comparison between sample fields predicted by the POU-MOR-Physics LES model and the fields from the filtered DNS. We note the close correspondence between the two sets of fields.

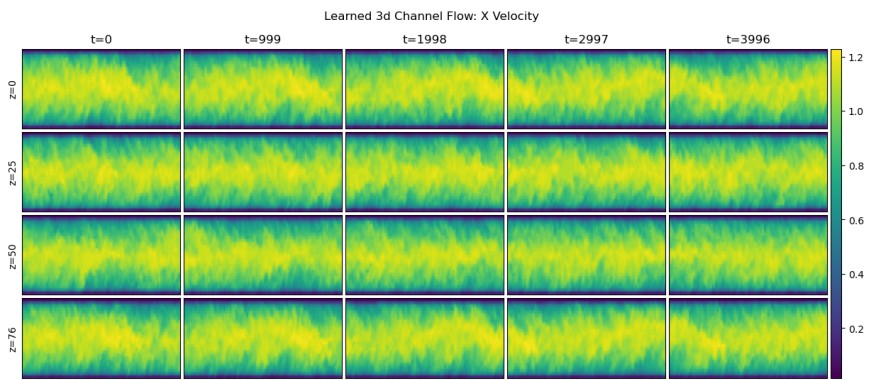

Figure 7: Learned Simulation: X Velocity

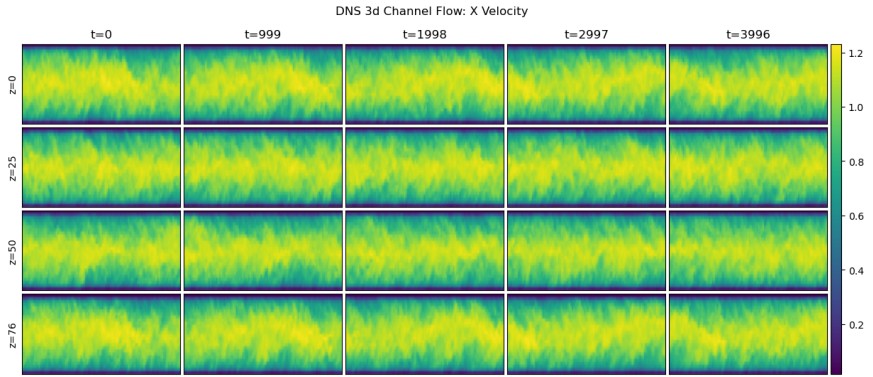

Figure 8: DNS: X Velocity

Figure 9: Comparison of X velocity field from learned simulation vs DNS from JHTDB.

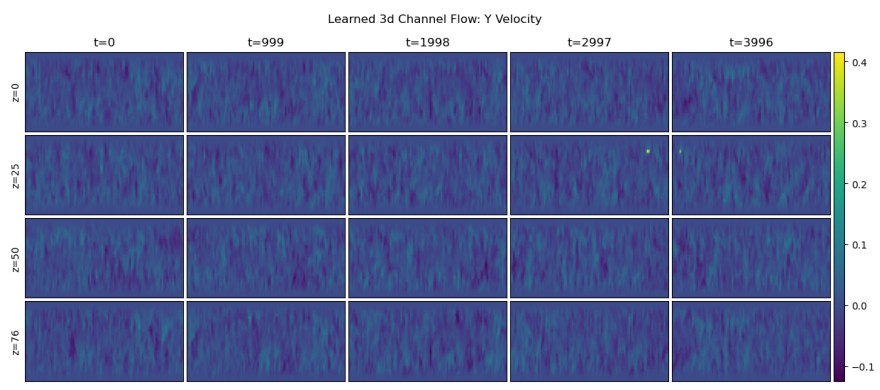

Figure 10: Learned Simulation: Y Velocity

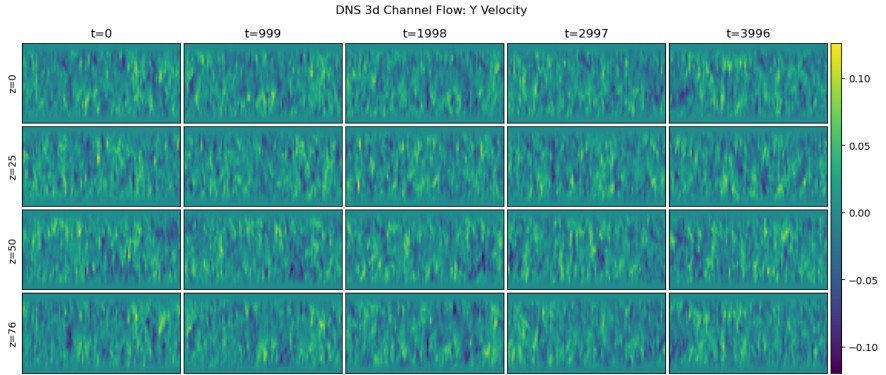

Figure 11: DNS: Y Velocity

Figure 12: Comparison of Y velocity field from learned simulation vs DNS from JHTDB.

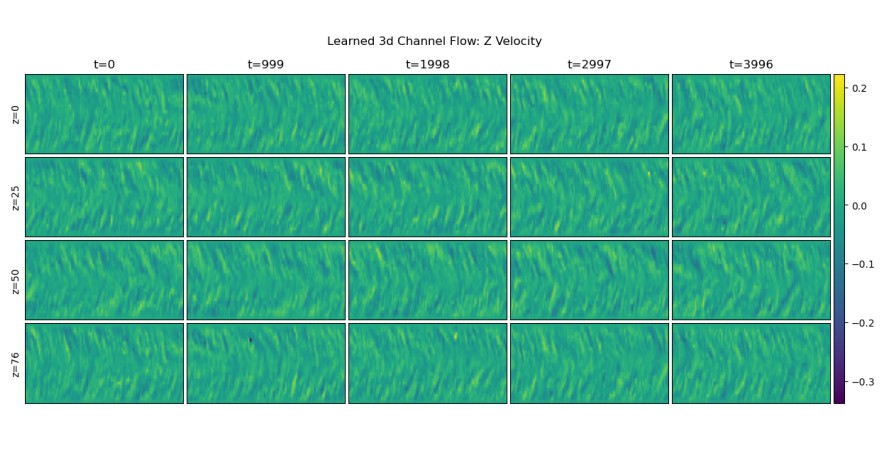

Figure 13: Learned Simulation: Y Velocity

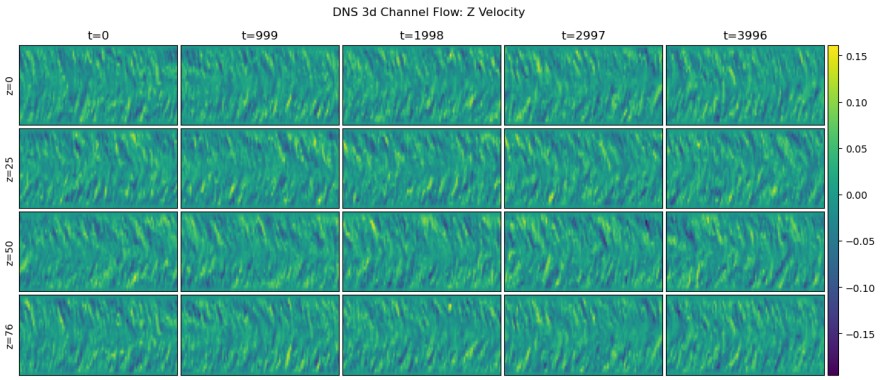

Figure 14: DNS: Y Velocity

Figure 15: Comparison of Y velocity field from learned simulation vs DNS from JHTDB.

## C  SMOOTH EXTENSION OF FUNCTIONS ON ARBITRARY DOMAINS TO THE TORUS

In this section, we provide details on our approach to smooth extension as discussed in section 2.1. We approach this problem by solving a constrained optimization problem that minimizes the $H^1$ semi-norm of the extended function such that it matches the original function with domain, $X$.

$$\min_{u_e} \quad \int \nabla u_e \cdot \nabla u_e dx$$
$$\text{s.t.} \quad \mathcal{R} u_e = u \tag{15}$$

The Lagrangian Bertsekas (2014) for this optimization problem is,

$$\mathcal{L} = \int \nabla u_e \cdot \nabla u_e dx + (\lambda, \mathcal{R} u_e - u)_{L_2(\mathcal{U})} \tag{16}$$

where the Lagrange multiplier, $\lambda \in U$. Using the Plancherel theorem, we rewrite the Lagrangian as,

$$\mathcal{L} = \sum_\kappa (\kappa \mathcal{F} u_e) \cdot (\kappa \mathcal{F} u_e) + (\lambda, \mathcal{R} u_e - u)_{L_2(\mathcal{U})} \tag{17}$$

Using the unitary Fourier transform, $\mathcal{F}^{-1} = \mathcal{F}^*$, the first order conditions give a linear system,

$$\nabla_{[u_e, \lambda]^T} \mathcal{L} = 0 \Rightarrow \begin{bmatrix} \mathcal{F}^{-1} \kappa \cdot \kappa \mathcal{F} & \mathcal{R}^T \\ \mathcal{R} & 0 \end{bmatrix} \begin{bmatrix} u_e \\ \lambda \end{bmatrix} = \begin{bmatrix} 0 \\ u \end{bmatrix}$$
$$= Aq = b \tag{18}$$

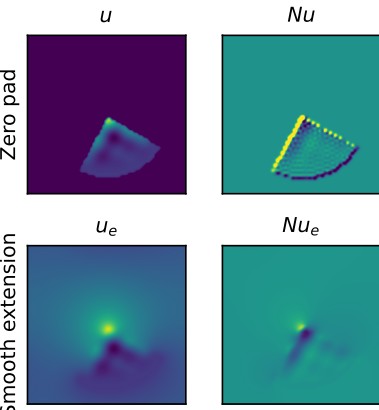

Figure 16: The smooth extension prevents Gibbs oscillation for Burgers action on a function originally defined on a quarter disk. *(Top right)* Simple extension of input function to torus via zero padding. *(Bottom right)* Smooth extension of input function to torus. *(Top left)* Burgers action on simple extension. *(Bottom left)* Burgers action on smooth extension.

where $\mathcal{R}^T$ is given by $(\mathcal{R}u_e, u)_X = (u_e, \mathcal{R}^T u)_{\mathbb{T}^d}$. We efficiently solve the normal equations for this system using matrix free operations in a conjugate gradient (CG) solver.

To illustrate this method, Figure 16 compares the smooth extension of an input function with an extension that sets the function values to zero outside of the domain and compares the action of the Burgers operator, $v = u \cdot \nabla u$, on the two extensions. The discontinuity in the simple extension leads to an oscillatory action while the action from the smooth extension is less oscillatory. This test was performed using input function data generated as per Section 3.1.

# D  IMPORTANCE NORMALIZED SOFTMAX

While our particular flavor of "mixture of experts" (aka partition of unity), is distinct from the name sake in Shazeer et al. (2017) we did find that our model benefited from a form of expert importance normalization inspired by Shazeer et al. (2017). In particular we used a Sinkhorn iteration Sinkhorn (1964) algorithm to normalize both total expert usage across space and the sum of gating weights at a particular point in space to 1.

$$(\tilde{\mathcal{S}}_{n+1}(\tilde{G})_i)(x) = (\tilde{\mathcal{S}}_n(\tilde{G})_i)(x) - \log(\int_{\tilde{x}\in\mathbb{T}^d} \exp(\tilde{\mathcal{S}}_n(\tilde{G})_i(\tilde{x}))d\tilde{x} - \log(\sum_{j=1}^{I}\exp(\tilde{\mathcal{S}}_n(\tilde{G})_j)(x)) \tag{19}$$

$$\text{s.t.} \quad \tilde{\mathcal{S}}_0(\tilde{G}) = \tilde{G} \tag{20}$$

$G = \mathcal{S}_n(\tilde{G}) = \exp(\tilde{\mathcal{S}}_n(\tilde{G}))$ is the gating weights resulting from importance normalized softmax with n Sinkhorn iterations (in our models n=10).

# E  DETAILS ON NONLINEAR POISSON EQUATION STUDY

This section provides training and archecture details on the problem examined in Section 3.1. To better isolate the effects of the mixture-of-experts model, we resort to a simpler POU-MOR-Physics model that only contains a mixture of single Fourier layers and removing the layer normalizations. We choose the gating network, $\tilde{G}$ to be a width 32 and depth 5 neural network. We use softmax to transform the gating network logits into weights. We choose the Fourier multipliers, $g$, to be neural network functions and $h$ and $g$ to be depth 5 neural networks with varying widths. We vary the widths of the latter two networks and target architectures with approximately (8000,16000,32000,64000,128000) total weights. We choose the unstacked variant of DeepONet for this study Lu et al. (2021), keeping the branch and truck networks fixed to a depth of 5 and varying the width to target the same above total parameter counts. We train all models using the Adam

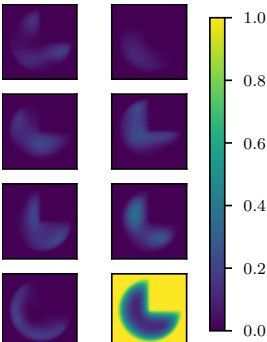

Figure 17: Partitions discovered by the POU-MOR-Physics operator in the left subplot of Figure 4

optimizer with the cosine learning rate scheduler with an initial learning rate of 0.01, $\alpha = 0.01$, and 10000 decay steps. We train all models with a batch size of 40 over 400 epochs.

In Figure 17, we show the partitions discovered by training the POU-MOR-Physics operator with 8 experts and $\sim 100K$ parameters.

## F  *a priori* KNOWN PHYSICS FOR LES MODEL

Our LES model is an autoregressive model with *a priori* known physics. See Section 2.2.3 for the general formulation of autoregressive models in our framework. In this section, we will start from the standard formulation of LES models and arrive at an LES model that incorporates a learned operator for various unclosed terms. As in Section 2.2.3, we will separate the known physics from the unknown and expose the unknown physics to operator learning.

Direct numerical simulation (DNS) solves the Navier-Stokes equations,

$$
\begin{aligned}
\partial_t v + \nabla v \otimes u &= -\frac{1}{\rho}\nabla p + \nu \nabla^2 v & x \in \Omega \\
\nabla \cdot v &= 0 & x \in \Omega \\
v &= 0 & x \in \partial\Omega
\end{aligned}
\tag{21}
$$

Equation equation 21 is typically expensive to integrate because real systems often contain a large range of spatiotemporal scales that must be resolved. The LES equations are obtained by applying a low pass spatiotemporal filter to the Navier-Stokes equations,

$$
\begin{aligned}
\partial_t \tilde{v} + \nabla \tilde{v} \otimes \tilde{v} &= -\frac{1}{\rho}\nabla \tilde{p} + \nu \nabla^2 \tilde{v} - \nabla \tau & x \in \Omega \\
\nabla \cdot \tilde{v} &= 0 & x \in \Omega \\
\tilde{v} &= 0 & x \in \partial\Omega
\end{aligned}
\tag{22}
$$

where $\tau = \widetilde{v \otimes v} - \tilde{v} \otimes \tilde{v}$ is the residual stress tensor. Since the small scale features have been filtered out, Equation equation 22, can be much cheaper to numerically integrate than Equation equation 21. However, $\tau$ is an unclosed term that must be modeled. Traditionally, one uses a combination of intuition and analysis to arrive at a simple model with a few parameters that can be fitted to DNS simulation or experiments. Here, we will use POU-MOR-Physics to provide the closure.

To obtain our model, we will first consider LES in a triply periodic domain.

$$
\begin{aligned}
\partial_t \tilde{v} + \nabla \tilde{v} \otimes \tilde{v} &= -\frac{1}{\rho}\nabla \tilde{p} + \nu \nabla^2 \tilde{v} - \nabla \tau & x \in \Omega \\
\nabla \cdot \tilde{v} &= 0 & x \in \Omega
\end{aligned}
\tag{23}
$$

We use the Chorin projection method to eliminate the pressure term and the explicit Euler time integrator to obtain an evolution operator,

$$
\begin{aligned}
\hat{v}^{n+1} =& \tilde{v}^n - \mathcal{F}^{-1}\left(i\kappa \cdot \mathcal{F}\tilde{v}^n \otimes \tilde{v}^n \right. \\
& \left. + \frac{i\kappa}{||\kappa||_2^2}(\kappa \otimes \kappa) : \mathcal{F}\tilde{v}^n \otimes \tilde{v}^n - ||\kappa||_2^2 \mathcal{F}\tilde{v}^n \right) \\
=& \mathrm{LES}(\tilde{v}^n)
\end{aligned}
\tag{24}
$$

where $\mathcal{F}$ is the Fourier transform and $\kappa$ is the wave vector.

This update operator does not include the boundary conditions, the missing sub-grid scale physics, or any forcing. We model these as an operator learned correction to the action above and obtain,

$$
\tilde{v}^{n+1} = \mathcal{P}(\mathrm{LES}(\tilde{v}^n)) = \mathcal{U}(\tilde{v}^n)
\tag{25}
$$

where $\mathcal{P}$ is the POU-MOR-Physics operator discussed in Section 2.2. We can learn the closure $\mathcal{P}$ from low pass filtered DNS data and obtain a model that operates on much lower dimensional features than the original DNS, as we demonstrate in Section 3.2 and Appendix B.

## G  POST PROCESSING DETAILS FOR JHTDB CHANNEL DATASET

Below in Table 4, we provide postprocessing details for the channel dataset.

| Parameter | Value |
|---|---|
| Spatial Stride | $s_x = s_y = s_z = 20$ |
| Sub-Sampled Dimensions | $103 \times 26 \times 77$ |
| Time Dimension | $t = 4000$ |
| Box Filter Dimension | $b = 20$ |

Table 4: Post Processing Parameters for JHTDB problem.

## H  TRAINING DETAILS FOR LARGE EDDY SIMULATION MODEL

Due to the memory demands of this learning task, we required parallelization to train our LES model. To handle the computational load efficiently, we employ 20 A100 GPUs alongside the Fully-Sharded Data Parallel (FSDP) strategy Zhao et al. (2023), thereby exploiting data-parallel training and necessitating a scaled learning rate that follows the batch size Goyal (2017).

To further mitigate the substantial memory requirements, we utilized the real Fast Fourier Transform (rFFT) within the MOR-Physics Operator Patel et al. (2021), effectively conserving memory without compromising performance.

We used the linear scaling rule from Goyal (2017) and the "One Cycle" warm-up schedule from Smith & Topin (2019); without warm-up, the model parameters may change too rapidly at the outset Goyal (2017) for effective training.

Adhering to the linear scaling rule from Goyal (2017), we set the batch size to batch_size $= n$ and the learning rate to learning_rate $= 1.25 \times 10^{-4} * n$, where $n$ represents the number of GPUs utilized. Although memory constraints necessitated a small per-GPU batch size of 1, we found that combining this approach with gradient clipping yielded effective training results.

To improve the stability of the learned subgrid dynamics operator, we employed several (i.e. 8) auto-regressive time steps during training, following methodologies similar to those proposed by Bengio et al. Bengio et al. (2015) (but without the curriculum). Statistical auto-regressive time series models often suffer from exponentially growing errors because the model's flawed outputs feed back into its inputs, compounding errors over multiple time steps. By exposing the model to this process during training — taking several auto-regressive steps before each optimization step — the model learns to correct its own compounding errors, mitigating problems at prediction time Bengio et al. (2015).

| Model: | ELBO: holdout ↑ | ELBO: 100 step ↑ | $R^2$: holdout ↑ | $R^2$: 100-step ↑ | MAE: holdout ↓ | MAE: 100 |
|---|---|---|---|---|---|---|
| Recursive Steps | 5.47153 | 5.2897 | 99.081% | 98.864% | 0.001802 | 0.0019609 |
| MoE+IN | 5.16139 | -93.94615 | 98.431% | 24.262% | 0.0026547 | 0.02711 |
| MoE | 5.06351 | -671.67914 | 98.505% | -699.483% | 0.0030291 | 0.080844 |
| Control | 5.12027 | -233.91377 | 98.743% | -488.797 % | 0.0029808 | 0.079651 |
| Ours | **5.63422** | **5.48693** | **0.99456%** | **0.99325%** | **0.0015041** | **0.0016127** |

Table 5: Ablation Study: Shorthand: IN=Importance Normalization, LN=LayerNorm, MoE=Mixture of Experts, PP-LL=Posterior-Predictive-Log-Likelihood.

## I   ABLATION STUDY AND SATO COMPARISONS

In our ablation study we choose to keep the amount of output (ground-truth) data fixed per step rather than the total amount of ground truth data per step across models. The reasoning here is two-fold: first of all in Bayesian theory the "weight" of the data (i.e. magnitude of the NLL) is proportional only to the size of the output data, second we derived scaling equations for both learning rate and gradient clipping keeping amount of output data constant represented a special case in these scaling equations where learning rate and gradient clipping could also be kept constant which seemed particularly elegant. Also the reason that the distinction between keeping total vs output amount of ground truth data per-step constant is meaningful is because we were comparing using recursive auto-regressive training vs 1-step-ahead training; in the former case model outputs are reused as model inputs which reduces the amount of ground-truth data needed in the inputs. We chose effectively to replace all model predictions in the inputs with actual data.

Also for all model *comparisons* we used "area resampling" (i.e. area *down*-sampling) to further reduce the 103x26x77 resolution of the production dataset described in subsection 3.2. "Area resampling" a continuous version of box averaging which is what we originally used to coarsen the much larger 2048×512×1536 DNS data to the 103x26x77 dataset in subsection 3.2 (which our production model uses). Specifically for the Ablation study and the CNN comparison we reduced the resolution to 70x17x52. For the WNO comparison we had to reduce the resolution to powers of 2: 64x16x64 (required by the WNO). Also all compared models as tested as LES models inside the forward Euler PDE solver we have. For all model comparisons in a particular bracket we approximated as best we could (given discrete architecture parameters) equal parameter count and used exactly equal training epochs as well as hyper-parameters like learning rate, gradient clipping, etc... Lastly for model comparisons we use the ELBO as the primary metric to determine the best model. This metric encompasses: uncertainty calibration and prediction accuracy. We calculate it for both: holdout data and long—100 step—horizon validation (and particularly decide the best model based on the long-horizon version). We also report $R^2$ and MAE of model mean predictions.

### I.0.1   HYPER-PARAMETERS

- Our large "single expert config" used in "Recursive Steps" and "Control" models used 6 layers and 36 hidden channels.

- While our mixture of experts config used 4 layers and 32 hidden channels per expert, and 4 point-wise convolution layers for the gating network.

- Also we used learning rate=0.0011253600000000002, and gradient clipping (by value) at 5.892556509887896. And the OneCycle Smith & Topin (2017) learning rate scheduler for all experiments.

The most obvious result of the ablation study is the importance of recursive multi-step-ahead training which really should be the only way to train an auto-regressive model like this. Since training 1-step-ahead then "transferring" to auto-regressive prediction represents distribution drift that hinders model performance. Similar results for auto-regressive language models have been shown in Bengio et al. (2015), however they use a curriculum which we don't. We aren't aware of this kind of result being established in auto-regressive timeseries field modeling or SciML in general however. It is also apparent that "importance normalization" was important for mixture of experts in this problem. It appears without it might over-rely on whichever expert is already the most accurate which leads

| Model: | ELBO: holdout ↑ | ELBO: 100 step ↑ | $R^2$: holdout ↑ | $R^2 : 100 - step$ ↑ | MAE: holdout ↓ | MAE: 100 step |
|---|---|---|---|---|---|---|
| WNO | 1.91353 | 2.16997 | -3.43538 | -1.78457 | 0.036095 | 0.031982 |
| Ours | **2.95503** | **2.82594** | **0.47534** | **0.34528** | **0.0172** | **0.018449** |

Table 6: WNO Comparison

| Model: | ELBO: holdout ↑ | ELBO: 100 step ↑ | $R^2$: holdout ↑ | $R^2 : 100 - step$ ↑ | MAE: holdout ↓ | MAE: 100 step |
|---|---|---|---|---|---|---|
| CNN-k4 | 3.41832 | 2.9057 | 0.24721 | 0.23969 | 0.022192 | 0.027813 |
| Ours-k4 | **3.54017** | **3.44403** | **0.37666** | **0.35351** | **0.017363** | **0.018602** |

Table 7: CNN Comparison

the model to under-utilize it's parameters. The mixture of experts method really is the best though when combined with recursive time stepping.

## I.1 WNO COMPARISON:

Wavelet Neural Operator Tripura & Chakraborty (2023a) was invented to alleviate the Gibbs phenomena weakness in FNO for complex non-periodic boundary conditions so it was a natural SATO comparison point. We found however the WNO was very slow to train (1/5 the speed of our model even after multiple optimizations), so we reduced the training epochs significantly (to just 150). We also used only 1 wavelet decomposition level because: it made it easier to match parameter count to our full-mode MOR Operator Mixture of Experts model and it doubled the training speed. We're not sure if the slow speed of WNO is algorithmic or just due to an inefficient implementation. Also since the WNO requires power of 2 resolutions we down-sampled to 64x16x64 as mentioned earlier. We trained special version of our model for the sake comparison because our model is best when using full-modes and our default problem has resolution 103x26x77 so we had to make a special one with 64x16x64 modes which also nearly matched the parameter count of the WNO. We didn't test the WNO with output Layer Normalization like we used for the CNN and our model simply because we wanted to test the architecture as-is. We actually beat the WNO quite squarely despite using a decent amount less parameters.

## I.2 CNN COMPARISON (CNO PROXY: OURS¿CNN¿=CNO):

CNO Raonic et al. (2023) is a SATO neural operator which attempts to alleviate aliasing errors impeding FNO **?** resolution invariance. These aliasing errors are caused by activations which introduce high frequency content that is aliased in the following Fourier transforms, so CNO solves this by following the activation functions by linear transformations and a low pass filter which return the "activations" to dense tensors which respect the band limit. It is effectively a CNN that operates in function space (via a form of interpolation) and adheres to a strict internal band limit to it's preserve resolution invariance property. It is of course following by a projection module like FNO has in order to recover some of the lost higher frequency information. It has been pointed out by reviewers of CNO however that while this innovation is advantageous–especially for resolution invariance–the imposed band limit could reduce expressive capacity to an extent. Our problem is a special case where we don't explicitly evaluate our model at different resolutions, so we figured we could use a CNN has a band-unlimited proxy comparison point for CNN since in the fixed resolution setting CNN¿=CNO, therefore if we establish that Ours¿CNN it follows that Ours¿CNN¿=CNO (for our turbulence problem, given fixed resolution). CNN also stands as a strong baseline comparison point on it's own since they remain the status-quo for SATO performance in general computer vision (which simulation surrogates fall under the umbrella of). We used a ResNet-style CNN, containing 10 layers (with kernel size: 4x4x4), skip connections and LayerNorm Ba et al. (2016) to normalize the activation statistics after summing the skip connections. We used LayerNorm because our batch size was very small due to large 3d video predictions and the GroupNorm paper **?** demonstrated that for ResNet with small batch sizes BatchNorm degrades far below LayerNorm and GroupNorm. However we tried GroupNorm and it didn't do as well as LayerNorm right away and we didn't have the budget to tune the number of groups to get better performance from GroupNorm. We could have used a U-Net architecture for comparison but that is out of scope.

Normally we use our model with maximum kernel size but the CNN wasn't capable of matching this since it is much slower at convolution $O(n^2)$ vs $O(nlog(n) + n) = O(nlog(n))$ (with kernel size k=n input resolution per dimension). So we matched the kernel size k=4x4x4 of the CNN in our model as well. This is actually significant limitation of the CNN though because our model can do much better by using the full kernel k=103x26x77 which is computational intractable for a CNN. This finding is also in contrast the common understanding about FNO's that the higher frequency modes are just noise and not very useful.

**Stability Features Introduced for CNN:**  We had to introduce two stabilizing features in order to make the CNN viable, it was very unstable in general—prone to diverging into NaNs during auto-regressive rollouts. These stabilizing features—while not strictly necessary for our models—were still observed to improve performance somewhat and *so for consistency we implemented them on our models as well*.

The stabilizing features were:

1. Bounding output mean predictions $\mu \leftarrow 2 * \tanh(\mu)$ (all values in the dataset $-0.5 < u < 1.5$ satisfy this bound).

2. Using LayerNorm directly on the model raw output "logits" $\mathcal{N}(u) = LayerNorm \circ x(u)$ (before application of $\mu$ and $\sigma$ transformations).

    (a) This may be counter-intuitive since it removes scale information and usually is reserved for hidden layers but the added stabilization was immensely helpful (especially given VI).

It should also be noted that we tried BatchNorm but encountered a pathological case where BatchNorm's different behavior between training and validation lead to catastrophic failure. This is because the model was an auto-regressive timeseries model so the same normalization layer was reused at different recursive time-steps where it encountered very different pre-normalization predictive distributions (e.g. imagine compounding error where early predictions are stable and later ones are erratic). Since it did actual normalization during training the model came to rely on it but during validation it simply became an affine transformation based on aggregate statistics from all time-steps it was used for which was not appropriate (causing NaN holdout performance). This is a unique situation where LayerNorm is actually categorically different and much more useful.

