# OpenReview forum: "Mixture of neural operator experts for learning boundary conditions and model selection"
_ICLR.cc/2026/Conference — ICLR 2026 Conference Withdrawn Submission_

### Official Review · Reviewer_HhLU · 2025-10-28

**Soundness:** 3
**Presentation:** 3
**Contribution:** 3
**Rating:** 6
**Confidence:** 3

**Summary:**

This paper proposes a Mixture-of-Neural-Operator Experts (MoNOE) framework that extends Fourier-based neural operators (FNO/MOR-Physics) to problems involving multiple physical regimes and non-periodic boundary conditions. The method uses a Partition-of-Unity gating network to blend several Fourier-based operator experts, each of which can specialize in a sub-region or physics regime. Known physics are embedded through a forward-Euler time-stepping operator, while unknown closures are learned by the neural operators. A Bayesian mean-field variational inference (MFVI) module provides uncertainty estimates. Applications include a nonlinear Poisson equation with mixed boundary conditions and a large-eddy-simulation (LES) closure model for turbulent channel flow. The approach achieves better accuracy than WaveletNO and CNN surrogates and yields interpretable domain partitions that align with flow regions.

**Strengths:**

- Several good ideas: MoE, adding explicit Euler update, importance normalized softmax

- Important problem: Tackles a key limitation of Fourier neural operators—their inability to handle heterogeneous domains, boundary conditions, and flow regimes—through a conceptually simple mixture-of-experts strategy.

- General utility: The MoE formulation could inspire many variants for other computational difficulties that require multiple specialized operators or adaptive refinement.

- Regularization of gating network: The use of Sinkhorn-style importance normalization to balance expert usage is well-motivated and practically effective.

- Integration of known physics: Embedding the explicit-Euler update provides a neat way to combine analytical physics and learned corrections in a differentiable pipeline.

- Demonstrated effectiveness: Results on Poisson and turbulent-flow benchmarks show clear accuracy gains and interpretable spatial partitions, while keeping the model relatively simple. LES simulations are good cases.

**Weaknesses:**

- Lack of theoretical guidance: The paper does not clarify when a Mixture-of-Experts architecture is necessary versus when a single FNO suffices. A theoretical or diagnostic criterion would make the contribution more prescriptive.

- No expert communication within a time step: Experts act independently and are only coupled implicitly through the evolving global field. This may limit physical consistency and flux continuity across expert boundaries.

- Limited theoretical analysis and comparisons: There is little discussion of computational cost or scalability of the smooth-extension step, and comparisons omit more recent operator-learning baselines.

- Clarity and focus. The manuscript is dense, and the main conceptual contributions could be emphasized earlier and more clearly.

- Figure 1 is identical to standard FNO. I can't tell the difference. Why not just the modified framework?

- Too much going on in the same paper: maybe move MFVI to the next paper?

- some missing latex references, e.g., line 466.

**Questions:**

- Do you have to use the euler update, or can you provide both the euler update and u_n?
- Are there ways to enable communication between experts in one step?
- Would enforcing flux or gradient continuity between experts (e.g., through overlap losses or shared latent states) improve physical consistency?
- Can we inform the NO of the boundary conditions as inputs?
- How sensitive are the learned partitions to changes in geometry or boundary conditions outside the training set?
- Does the MFVI uncertainty correlate with physical errors (e.g., turbulence statistics), or is it mainly a calibration tool?
- How costly is the smooth-extension step in practice, and could it be replaced by a learned or implicit periodic embedding?

---

### Official Review · Reviewer_qzzE · 2025-10-30

**Soundness:** 3
**Presentation:** 2
**Contribution:** 2
**Rating:** 6
**Confidence:** 4

**Summary:**

The paper combines mixture of expert (MOE) techniques with operator learning for boundary condition in partial differential equations. It applies domain decomposition (partition of unity) via MOE, which is inspired by "volume penalized" methods. Experiments with the JHU turbulence dataset show that the proposed method outperform WaveletNO and CNN.

**Strengths:**

- The paper studies an interesting and hard problem of 3D tuburlence with boundary.
- The paper combines MoE with neural operators.
- The numerical results are encouraging.

**Weaknesses:**

Result:
- According to Table 1, the recursive step (first row) already has decently error rates, does it suggest the MoE in the model is less significant?
- It is great that the author considers 3D turbulence dataset. But currently it is only compared against WNO. Ideally there could be more baselines.

Writing:
- The Volume Penalization method is a bit vague. It will be better to define it more rigoriously.
- Sec 2.3 seems not very connected to the main paper. It would be better to motivate why MFVI is needed.
- Figures should be larger.
- in equation (3), it would be better to avoid abusing notation P
- it will be great to report the resolution (grid size), runtime, and memory usage.

**Questions:**

Method:
- It seems in the current form, all Fourier layers in MoE are defined on the full domain, just with different weights. To avoid Gibbs phenomena ideally one want define Fourier series on different domains, right?
- The standard MOE is done on the nonlinear layer h. It would be interesting to add an ablation whether it is better to add MoE on mlp (h) or ourier layer (N)

Related works:
- A similar gating mechanism has been explore in GNOT [1]. It will be interesting to discuss and compare.
- Regarding Gibbs phenomena, [2] studied extended domain for Fourier transform.

[1] Hao, Zhongkai, et al. "Gnot: A general neural operator transformer for operator learning." International Conference on Machine Learning. PMLR, 2023.
[2] Ganeshram, Adarsh, et al. "FC-PINO: High Precision Physics-Informed Neural Operators via Fourier Continuation." arXiv e-prints (2022): arXiv-2211.

---

### Official Review · Reviewer_LXXi · 2025-10-30

**Soundness:** 3
**Presentation:** 1
**Contribution:** 2
**Rating:** 4
**Confidence:** 4

**Summary:**

This work propose to use mixture-of-expert to efficiently learn different neural operators in different flow regions. It also integrates a Bayesian mean-field variational inference (MFVI) approach for uncertainty quantification and autoregressive training for temporal evolution. The paper demonstrates improvements over DeepONet and WNO on 3D turbulent channel flow using filter DNS data.

**Strengths:**

1. The main experiment shows improvement on JHTDB, an important 3D dataset that is generally missing in current works.
2. The spatial partitions learned by the gating network correspond to meaningful physical subregions, e.g., specific experts are assigned to near-wall regions.
3. The authors shows (Figure 4) interesting scaling ability when adding more experts.

**Weaknesses:**

Presentation suggestions:
1. A few labels references are undefined. The paper format in general needs a careful check.
2. Transitions in Sec.2 are hard, especially when jumping from MOR to MFVI. Sec. 2.3 is dense and hard to follow.
3. The quality of figures and tables generally need improvement. Some figures are small and hard to read.

Experiments:
1. A direct comparison to Wang 2024 (IUFNO) on JHTDB is missing.
2. The geometric flexibility of proposed method is only demonstrated on Poisson.

**Questions:**

1. Why the gating network is not designed take velocity as input? It's also natural for the network to determine expert assignment based on flow conditions, etc.
2. The expert assignment in channel flow seems to be very regular and can be replaced by a manually designed map in prior, e.g., the thickness of the boundary layer. I acknowledge that the improvements might come from both the way that experts are assigned, and helping the training. Can you justify this?

---

### Official Review · Reviewer_3oUL · 2025-11-03

**Soundness:** 3
**Presentation:** 1
**Contribution:** 2
**Rating:** 2
**Confidence:** 4

**Summary:**

The paper proposes their method to solve the challenges that existing neural operator works are facing: the problems with challenging boundary conditions. The proposed method uses mixture of experts to have experts adptively focus on their domain. Another contribution is that the proposed method uses variational inference to achieve posterior estimation.

**Strengths:**

1. This paper extensively utilized multiple techniques for learning the non-periodic boundary condition and discontinuities in PDE problems, which includes a mixture of experts, domain decomposition, and variational inference.
2. The experiments show that the proposed method beat all of the baselines. Thus this paper empirically shows the advantages and soundness of the proposed method.

**Weaknesses:**

1. The presentation needs lots of improvements.
2. Line 466: “See Appendix ?? for details…”
3. The font size across figures are inconsistent and some of them are too small.
4. The font type is inconsistent across figures.
5. The proposed method is a combination of several techniques. Thus, the novelty of the proposed method is not enough.
6. Formatting issue: too much blank space in page 12. The table 5 is cut off in page 19.
7. For the experiments, regarding the main results, the authors only compare with one or two baselines. To fully justify the proposed methods, more baselines should be considered.

**Questions:**

1. Why put Table 1 in the main text if you put the ablation studies in the appendix, and where are Table 1,2,3 referenced?

---

### Note · Authors · 2025-12-02

**Comment:**

Thanks for the reviews. We have decided to withdraw this submission and continue working to improve our manuscript.

**Withdrawal Confirmation:**

I have read and agree with the venue's withdrawal policy on behalf of myself and my co-authors.